# CoMMIT: Coordinated Instruction Tuning for Multimodal Large Language Models

## Abstract

Instruction tuning in multimodal large language models (MLLMs) generally involves smooth integration of a backbone LLM and a feature encoder that has non-text input modalities. The major challenge is how to efficiently find the synergy through cooperative learning, so that LLMs can adapt their reasoning abilities in downstream tasks while feature encoders can adjust to provide more task-specific information of its modality. In this paper, we analyze the MLLM instruction tuning from both theoretical and empirical perspectives, where we find unbalanced learning between the two modules, i.e., the feature encoder and the LLM, can cause problems of oscillation learning and insufficient training with diminishing learning gradients. Inspired by our findings, we propose a Multimodal Balance Coefficient that enables quantitative measurement on the balance of learning. Based on this, we further design a dynamic learning scheduler that better coordinates the learning between the LLM and feature encoder, alleviating the oscillation and insufficient training. In addition, we introduce an auxiliary regularization on the gradient to promote updating with larger step sizes, which potentially allows for a more accurate estimation of the proposed MultiModal Balance Coefficient and further improves the training sufficiency. Our techniques are agnostic to the architecture of LLM and feature encoder, so can be generically integrated with various MLLMs. We conduct experiment results on multiple downstream tasks with various MLLMs, demonstrating the proposed method is more effective than the baselines in MLLM instruction tuning.

## 1 Introduction

Multimodal instruction tuning aligns pre-trained multimodal large language models (MLLMs) with specific downstream tasks by fine-tuning MLLMs to follow arbitrary instructions (Dai et al., 2024; Zhang et al., 2023; Zhao et al., 2024; Lu et al., 2023; Han et al., 2023; Wu et al., 2024a; Wang et al., 2024b). State-of-the-art pre-trained MLLMs (Li et al., 2023; Liu et al., 2024; Tang et al., 2023a; Chu et al., 2023) generally adopt a similar model architecture design. Specifically, the non-text data (image, audio, etc) is first encoded by a feature encoder into embedding tokens. Then, these encoded embeddings are inserted into language prompts, consisting of the multimodal sequences as input to an LLM. Multimodal understanding and reasoning with MLLMs relies on the

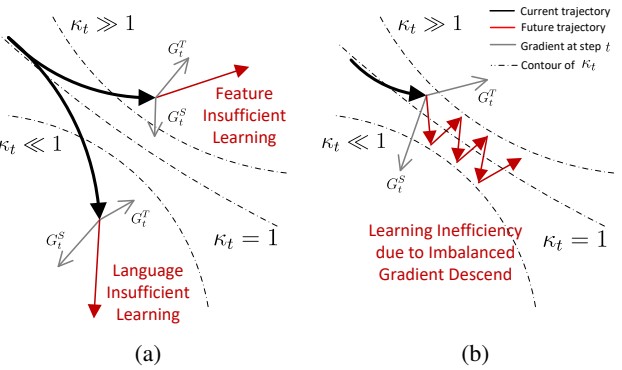

Figure 1: Illustration of (a) single modality learning insufficiency problem, (b) and multimodal learning oscillation problem, caused by imbalanced multimodal learning. We show the optimization trajectories in solid bold lines and the multimodal gradients at the current step $t$ in solid thin lines. The dashed line borders are the contours of the learning balance coefficient $\kappa_t$ proposed and detailed in Section 4.

ability to learn aligned multimodal features with the feature encoder (*e.g.*, Li et al. (2023)) and the

pre-trained abilities of the backbone LLMs (*e.g.*, Touvron et al. (2023); Chiang et al. (2023)) to understand the multimodal input. In the instruction tuning of these MLLMs, it is critical to cooperatively learning and align the feature encoder and the backbone LLM. The challenge lies in two folds: (1) the encoded non-text (*e.g.*, vision and audio) features in downstream tasks might not be perfectly aligned with the text features, which requires the backbone LLM to adjust its pre-train parameters to recognize the new feature tokens from non-text modalities; (2) While LLMs are already knowledgeable of different reasoning tasks in their pre-trained, the feature encoders require adjustments to provide more relevant modal-specific information to the downstream tasks. As a result, the insufficiently trained LLMs in (1) that fail to comprehend the non-text modalities can suffer from hallucination problems (Bai et al., 2024; Rawte et al., 2024), due to the strong language prior in backbone LLMs. On the other hand, insufficient learning of the feature encoder in (2) may cause information loss (Bai et al., 2024; Tong et al., 2024), resulting in inadequate evidence for LLM reasoning. Therefore, it is essential to balance the learning between the feature encoder and backbone LLM, so that the learning is not overly biased on either of the two modules.

In this paper, we first propose a multimodal balance coefficient that quantifies the balance of learning between the feature encoder and the backbone LLM in MLLM instruction tuning. Based on theoretical analysis and empirical observations, we identify two types of learning dilemmas caused by imbalanced learning in multimodal instruction tuning: *i)* the insufficient learning problem and *ii)* the oscillation problem, which are illustrated in Figure 1 and can be describe by our proposed multimodal balance coefficient. Specifically, in Figure 1a shows the insufficient learning problem where the training is largely inclining towars either the LLM or the feature encoder (or overfit to one of the two modules). In such cases, the effective gradient descent is mostly only updating the LLM or the feature encoder, resulting in insufficient learning of the other module. This makes the gradient descent less effective, since the insufficiently learned modult (LLM or feature encoder) cannot contribute sufficient information to the output generation. The other learning difficulty in Figure 1b, is a demonstration of the oscillation problem, which happens when the learning is alternatively inclining toward either the feature encoder or the LLM. This will impede the convergence of the optimization process and undermine learning efficiency.

To address these issues, we propose **Co**ordinated **M**ulti**M**odal **I**nstruction **T**uning (**CoMMIT**), which consists of a coordinated learning rate scheduler (Section 6.1) and regularization in gradient descent (Section 6.2). The coordinated learning rate scheduler dynamically adjust the learning rate of the feature encoder and LLM according to multimodal balance coefficient, which avoids the inefficiency caused the oscillation problem while allowing sufficient gradient descent for both of the two modules. The regularization promotes updates with larger step sizes during training. This alleviates the gradient diminishing problem and further reduce the chance of insufficient training. In addition, we theoretically analyze the convergence rate and demonstrate that we can achieve faster convergence when optimizing with **CoMMIT** (Section 7). We summarize our main contributions as follows:

- We introduce a theoretical framework to uncover the pitfall of the learning imbalance problem in MLLM instruction tuning, which can cause MLLM insufficient learning and the oscillation problem.

- Based on the theoretical analysis and empirical observation, we propose **CoMMIT** to balance multimodal learning progress by dynamically coordinating learning rates on the feature encoder and LLM. **CoMMIT** also enforce a gradient regularization that encourage larger step sized and further avoid infufficient training.

- Applying **CoMMIT** introduces a novel term in the convergence rate analysis. Theoretical analysis proves that this term is always greater than one, leads to faster convergence. We also demonstrate that the theorem can be generalized across various optimizers.

- Empirical results on multiple downstream tasks in vision and audio modalities with various LLM backbones show the efficiency and effectiveness of the proposed methods. We demonstrate that **CoMMIT** can better coordinate multimodal learning progress and reduce learning oscillations.

## 2 RELATED WORKS

MLLMs have become a new paradigm to empower multimodal learning with advanced language reasoning capabilities, such as with vision (Li et al., 2023; Liu et al., 2024; Wang et al., 2024c; Maaz et al., 2023; Zhang et al., 2023; Huang et al., 2023a; Yan et al., 2024), and audio (Huang et al., 2023b; Tang et al., 2023a; Gardner et al., 2023). Despite good generalizability and zero-shot performance of existing large language models (LLMs), the discrepancy between different modalities can be one of the greatest challenges for LMMs to achieve comparable reasoning performance as LLMs. To bridge the multimodality gap and align with downstream tasks, several works focus on two-fold considerations: feature (modality) alignment and reasoning alignment. The most common approach for feature alignment is to encode the source modality feature to semantic tokens within the LLMs' embedding feature space. By adding the modality-specific tokens (Wang et al., 2024a; Liu et al., 2021a; Zhang et al., 2024) as soft prompt inputs (Liu et al., 2021b; Xie et al., 2023; Wu et al., 2023; 2024b), the backbone LLMs can process these tokens with language tokens as a unified sequence. However, the newly added semantic tokens cannot be understood by LLMs directly for language reasoning, due to the limited text-only pretraining of LLMs. Such misalignment problems will lead to textual hallucination problems, namely *linguistic bias* (Ko et al., 2023; Tang et al., 2023b), in which the language models reason only based on their language prior. Thus, multimodal alignment should be achieved by additional adaptation of the LLM itself with multimodal instruction tuning.

## 3 PRELIMINARIES: INSTRUCTION TUNING WITH MLLM

Given a pair of non-text input $I^S$ (images, audio, etc) and instruction prompt $I^X$ of nature language, the instruction tuned MLLM should comprehend the semantics of $I^S$ and generate outputs that comply with the instruction specified in $I^X$. State-of-the-art MLLM instruction tuning generally adopt a similar diagram of training(Gardner et al., 2023; Li et al., 2023; Liu et al., 2024), which involves cooperative training between a feature encoder $S$ and a pretrained language model $X$. Specifically, $S$ first encodes the multimedia input $I^S$ into the embedding space of $X$. Then, the encoded $I^S$ is inserted into the instruction prompt $I^X$ as input that conditions the output generation with $X$,

$$P_{S,X}(\hat{y}_k|I^S, I^X, y_{j<k}) = X([S(I^S), I^X, y_{j<k}]),$$

where $y_{j<k}$ are the expected ground truth tokens before the $k$th predicted token $\hat{y}_k$ in auto-regressive generation, $k = 1, \cdots, K$. The training loss is the cross-entropy defined on the predicted distribution on $\hat{y}_k$ and the ground truth $y_k$(Liu et al., 2024; Ouyang et al., 2022),

$$L\left(Y = \{y_k\}_{k=1}^K \mid I_S, I^T\right) = -\frac{1}{K}\sum_{k=1}^K y_k \log P_{S,X}(\hat{y}_k|I^S, I^X, y_{j<k}), \quad (1)$$

The learning objective is to find the optimal $X$ and $S$ by minimizing the loss function. As mentioned in Section 1, the training of MLLM can either be insufficient by biasing toward only one of $S$ and $X$, or inefficient by oscillating between the optimization of the two modules. Our goal is to find a balance between the learning of $X$ and $S$, so to accelerate the convergence of training while ensure that both $S$ and $X$ are learned with suffucient knowledge.

## 4 MEASUREMENT OF LEARNING BALANCE IN MLLM INSTRUCTION TUNING

To assess the balance between the updates on $X$ and $S$, we first measure the significance of each update separately with $X$ and $S$. Formally, for the $t$-th step of training, we define $d(P_{X_t,S_t}||P_{X_{t+1},S_t})$ and $d(P_{X_t,S_t}||P_{X_t,S_{t+1}})$ that quantify the significance of updates on $X$ and $S$, respectively, by measuring the shift in output distribution,

$$d(P_{X_t,S_t}||P_{X_{t+1},S_t}) = \frac{1}{K}\sum_k \mathbb{KL}\left(P_{S_t,X_t}(\hat{y}_k|I^S, I^X)||P_{S_{t+1},X_t}(\hat{y}_k|I^S, I^X)\right), \quad (2)$$

$$d(P_{X_t,S_t}||P_{X_t,S_{t+1}}) = \frac{1}{K}\sum_k \mathbb{KL}\left(P_{S_t,X_t}(\hat{y}_k|I^S, I^X)||P_{S_t,X_{t+1}}(\hat{y}_k|I^S, I^X)\right) \quad (3)$$

where we use the subscript $t$ to index the trained steps and $\mathbb{KL}(\cdot||\cdot)$ is the KL divergence. Based on Eq.2 and 3, we define the Multimodal Balance Coefficient that meaures the balance between training on $X$ and $S$.

**Definition 4.1** (Multimodal balance coefficient). *For time step $t$ of joint training on $X$ and $S$, the Multimodal Balance Coefficient $\kappa_t$ is measured considering the separate learning steps on the feature encoder $S_t$ and the language model $X_t$.*

$$\kappa_t = \frac{d(P_{X_t,S_t}||P_{X_{t+1},S_t})}{d(P_{X_t,S_t}||P_{X_t,S_{t+1}})}. \tag{4}$$

$\kappa_t >> 1$ and $\kappa_t \to 0$ indicates that the learning is inclining toward $X$ and $S$, respectively. $\kappa_t$ with high variance corresponds to learning that oscillates between optimizing on $X$ or $S$. To further illustrate this, we derive the Theorem 1 that estimated the gradient on $X$ and $S$ during training.

**Theorem 1.** Let $G_t^X$ and $G_t^S$ be the gradient on $X$ and $S$ at time step $t$, we can derive the multimodal gradient estimated bounds,

$$\|G_t^X\| \le (\kappa_t + 1)H_t^S, \quad \|G_t^S\| \le (\frac{1}{\kappa_t} + 1)H_t^X, \tag{5}$$

where $H_t^S$ and $H_t^T$ represent the individual learning steps of $S$ and $X$, respectively. These are given by,

$$H_t^S = \left(\|I_t^X\| + \|S_t(I_t^S)\|\right)^{-1} \|logits(P_{X_t,S_{t+1}})\|,$$
$$H_t^X = \left\|I_t^S\right\|^{-1} \|logits(P_{X_{t+1},S_t})\|. \tag{6}$$

The detailed proof are provided in Appendix A.1.

Within the metric space of generated probability distribution $(P_{S,X}, d)$, the value of $H_t^S$ and $H_t^T$ are bounded by a finite norm. $H_t^S$ and $H_t^T$ are also lower-bounded, assuming multimodal gradients are not diminishing which we alleviate by proposing a regularization on gradient in Section 6.2. Therefore, $\kappa_t$ can account the most for the gradient upper bound in Eq. 5 so is suitable to measure the learning imbalance problem.

**Learning dilemmas in MLLM instruction tuning.** Based on the definiation and analysis above, we propose two hypotheses regarding potential learning dilemmas in MLLM instruction tuning, which are further evaluated in Section 5 and 8.

**Hypothesis 4.1** (Learning inefficiency). *The oscillation of the multimodal learning balance coefficient $\kappa_t$ can cause an inefficient learning problem that slows the training and convergence.*

**Hypothesis 4.2** (Learning insufficiency). *When $\kappa_t >> 1$ or $\kappa_t \to 0$, the imbalanced learning that inclines toward either $X$ or $S$ can cause the insufficient learning problem.*

In Section 5, we observe the dynamics of $\kappa_t$ is different experiment settings. Based on the observation, we propose CoMMIT in Section 6 which alleviates the identified learning insufficiency and inefficiency problems. Specifically, CoMMIT consists of a Coordinated Learning Rate Scheduling (Section 6.1) that strikes a balance between training on $X$ and $S$, along with a regularization loss (Section 6.2) that avoids gradient diminishing and further promotes sufficient training.

## 5 EMPIRICAL STUDY OF LEARNING DILEMMAS IN MLLM INSTRUCTION TUNING

We conduct an empirical study of MLLM instruction tuning to understand the behavior of $X$ and $S$ in multimodal instruction tuning. The experiment is conducted on a visual question-answering task TextVQA (Singh et al., 2019), on which a BLIP-2 (Li et al., 2023) model is instruction-tuned. We show the analysis results on TextVQA, one of the common instruction tuning downstream tasks which is widely used in vision LLMs (Dai et al., 2024; Yin et al., 2024). To probe the problem of imbalance and insufficient learning, we include three learning strategies: (1) **Synced LR** is trained by setting the learning rate of both $X$ and $S$ to $1e-4$; (2) **Language LR** $\uparrow$ increases learning rate of $X$ to $1e-3$; (2) **Encoder LR** $\uparrow$ increases the learning rate of $S$ to $1e-3$.

### 5.1 The Oscillation Problem in Imbalanced MLLM Instruction Tuning

To quantitatively understand the effect of the imbalanced multimodal learning problem in MLLM instruction tuning, we show the learning curves (Figure 2) of the measurement variables $H_t^S$, $H_t^X$, and $\kappa_t$ proposed in Eq. 5.

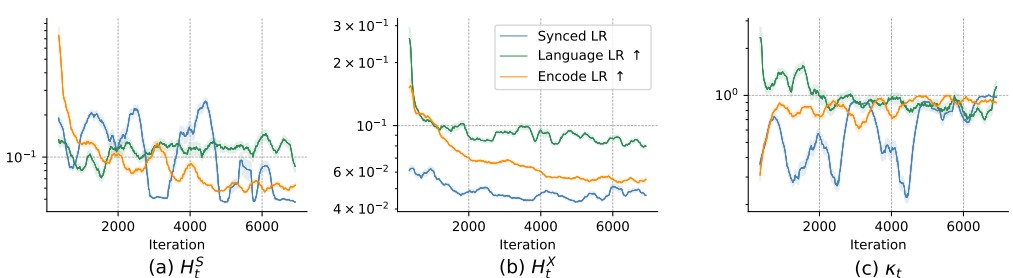

(a) $H_t^S$     (b) $H_t^X$     (c) $\kappa_t$

Figure 2: Learning curves of the variables $H_t^S$, $H_t^X$, and $\kappa_t$ for a measurement of learning balance in BLIP-2 instruction tuning on TextVQA.

**Observation 5.1.** As shown in Figure 2(c), the multimodal learning process can suffer from significant oscillation problems in the **Synced LR** setting.

Specifically, the learning curve of $\kappa_t$ in the **Synced LR** setting varies around the value of 1 (*i.e.*, an absolute balance), which is a showcase of the oscillation problems that signify training instability. Interestingly, it can be found that the feature encoder $S$ is more unstable than the language model $X$, by comparing $H_t^S$ in Figure 2(a) and $H_t^X$ in Figure 2(b). By increasing the learning rate either $X$ (**Encoder LR**) or $S$ (**Language LR**), we can observe that the three metrics in Figure 2 are stabilized. In the next section, we show that such a stabilization is at the expense of insufficient training. We further demonstrate Hypothesis 4.1 that oscillation can cause inefficient training in Section 8.

### 5.2 The Learning Insufficiency in Imbalanced MLLM Instruction Tuning

Let $\theta_t^S$ and $\theta_t^X$ be the parameters of $S$ and $X$ at time step $t$. We further show three metrics with the same backbone MLLM and training data as in Section 5.1: (1) the normalized learning gradient $\|G_t^S\|/\|\theta_t^S\|$ of the feature encoder $S$ in Figure 3(a), (2) the normalized learning gradient $\|G_t^X\|/\|\theta_t^X\|$ of the language model $X$ in Figure 3(b), and (3) the cross-entropy loss in Figure 3(c), to understand the impact of imbalanced learning between $X$ and $S$ on the learning sufficiency in MLLM instruction tuning.

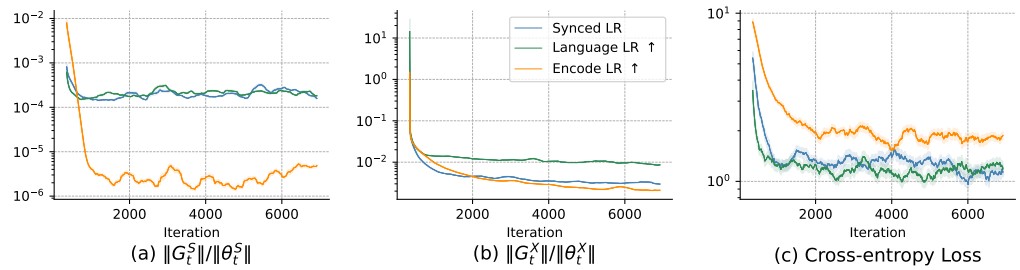

(a) $\|G_t^S\|/\|\theta_t^S\|$     (b) $\|G_t^X\|/\|\theta_t^X\|$     (c) Cross-entropy Loss

Figure 3: The learning curves of normalized learning gradient $\|G_t^S\|/\|\theta_t^S\|$ and $\|G_t^X\|/\|\theta_t^X\|$ for the feature encoder and language model respectively, as well as the cross-entropy training losses.

**Observation 5.2.** In Figure 3(c), we observe that imbalanced learning that inclines toward $X$ or $S$ (*e.g.*, **Encoder LR** ↑) can slow the convergence of the MLLM with gradient diminishing and inferior training performance.

This is consistent with Hypothesis 4.2, since the diminishing gradient would result in insufficient training with gradient descent. For example, we can observe in Figure 3(a) and Figure 3(b) that

the **Encoder LR** can simultaneously cause the gradient diminishing in both $X$ and $S$, with the cross-entropy converging to a higher value in Figure 3(c). In such cases, it is necessary to strategically balance the learning between different modules, so the training is not inclining toward either $X$ or $S$.

# 6 CoMMIT: COORDINATED MULTIMODAL INSTRUCTION TUNING

## 6.1 COORDINATED LEARNING RATE SCHEDULING

Based on the observations in Section 5, the learning rate on $X$ should be boosted when the training is inclining toward $S$ ($\kappa_t \to 0$), and vise verse. So motivated, we propose a dynamic learning rate scheduling method to coordinate multimodal learning between $X$ and $S$, which alleviates the learning oscillation problems while enabling sufficient training for both $X$ and $S$.

Inspired by damping strategies in optimization (Lucas et al., 2018; Tanaka & Kunin, 2021; Wei et al., 2021), we use the proposed learning balance metric $\kappa_t$ in Eq. 4 as the damping parameter that facilitates a balanced multimodel learning. Specifically, we track the $N_\kappa$ moving average of $\kappa_t$ through the learning process,

$$\tilde{\kappa}_t = \frac{1}{N_\kappa} \sum_{i=1}^{N_\kappa} \kappa_{t-i+1}, \tag{7}$$

then dynamically adjust learning rates of $X$ and $S$ in accordance. Let $\beta_t^X$ and $\beta_t^S$ be the learning rates on $X$ and $S$ at time step $t$. We adjust the learning rates by,

$$\beta_t^T = \frac{2\alpha}{\tilde{\kappa}_t + 1}, \ \beta_t^S = \frac{2\alpha}{1/\tilde{\kappa}_t + 1}, \tag{8}$$

where $\alpha$ is the base learning rate. To avoid a large computation overhead for batch-wise calculation of $\tilde{\kappa}_t$ and reduce the noise caused by frequent adjustment of the learning rates, we only periodically update the learning rates for every $N_{lr}$ time steps.

## 6.2 REGULARIZATION

During training, the diminishing $H_t^S$ and $H_t^T$ as observed in Figure 3 can cause higher estimation errors in $\tilde{\kappa}_t$. To address these, we propose an auxiliary regularization that encourages large step sizes for both $X$ and $S$, which mitigates gradient diminishing. Specifically, we want to encourage larger distribution drifts $d(P_{X_t,S_t}||P_{X_{t+1},S_t})$ for $X$ and $d(P_{X_t,S_t}||P_{X_t,S_{t+1}})$ for $S$, apart from gradient descending on the cross-entropy loss Eq. 1.

The gradient update for our proposed CoMMIT at time step $t$ is,

$$\theta_{t+1}^X \leftarrow \theta_t^X - \beta_t^X \cdot \nabla_{\theta^X} L(S_t(\tilde{X}_{\theta_t^X})) + \beta_t^X \cdot \nabla_{\theta^X} d(P_{X_t,S_t}||P_{X_{t+1},S_t}), \tag{9}$$

$$\theta_{t+1}^S \leftarrow \theta_t^S - \beta_t^S \cdot \nabla_{\theta^S} L(T_t(S_t(X)); \theta_t^S) + \beta_t^S \cdot \nabla_{\theta^S} d(P_{X_t,S_t}||P_{X_t,S_{t+1}}). \tag{10}$$

Note that the distribution drifts $d(P_{X_t,S_t}||P_{X_{t+1},S_t})$ and $d(P_{X_t,S_t}||P_{X_t,S_{t+1}})$ does not involve ground truth labels. Therefore, our proposed regularization is potentially also applicable to unsupervised instruction tuning.

# 7 THEORETICAL ANALYSIS

In this section, we present the computation and proof of a new convergence bound with our proposed method. Our theoretical analysis demonstrates that it achieves a faster convergence rate compared to the imbalanced MLLM instruction tuning.

## 7.1 SETUP AND NOTATIONS

Consider a non-convex random objective function $F : \mathbb{R}^d \to \mathbb{R}$. In the context of large-scale optimization, this function can be effectively expressed as the average of $N$ component functions, denoted as, $F(x) = \frac{1}{K} \sum_{k=1}^{K} f_k(x)$, where each $f_k(x)$ is an i.i.d sample. We are going to minimize the expect value of $\mathbb{E}[F(x)]$ given $x \in \mathbb{R}^d$. We also define $\mathbb{E}_{k-1}$ as the conditional expectation with

respect to $f_1, f_2, \cdots, f_k$. Similar as Adam (Kingma & Ba, 2014) algorithm, we denote $m_{k,i}, v_{k,i}$, $x_{k,i}$ as the $i$-th component of $m_k, v_k, x_k \in \mathbb{R}^d$ iteratively. Building upon the insight of Défossez et al. (Défossez et al., 2020) regarding the presence of two bias correction terms $m_k$ and $v_k$, we define $\alpha_{k,i} = \alpha_i \sqrt{\frac{1-\beta_2^k}{1-\beta_2}}$. Notably, we opt to drop the correction term for $m_k$ due to its faster convergence compared to $v_k$.

Aligned with our proposed methodology, we incorporate two additional terms into the original Adam algorithm. A dynamic learning parameter $\lambda$ that balances feature and language learning is designed to adapt based on changes in $\tilde{\kappa}$. To mitigate the risk of vanishing or exploding gradients, we introduce an auxiliary loss regularization function $h(x)$, defined in Section 6.2 to enhance training stability and support the overall robustness of the learning process. By setting $\beta_1 = 0$, $0 < \beta_2 \leq 1$, $\alpha_{k,i} > 0$, $\epsilon = 10^{-8}$, $m_0 = 0$, and $v_0 = 0$, given $x_0 \in \mathbb{R}^d$ as our starting point, this refinement yields the updated rules as follows,

$$v_{k,i} = \beta_2 v_{k-1,i} + (\lambda \nabla_i f_k(x_{k-1}) + \nabla_i h_k(x_{k-1}))^2 \tag{11}$$

$$x_{k,i} = x_{k-1,i} - \alpha_k \frac{\lambda \nabla_i f_k(x_{k-1}) + \nabla_i h_k(x_{k-1})}{\sqrt{v_{k,i} + \epsilon}} \tag{12}$$

Throughout the proof, we also assume the norm of the gradients $\|\nabla f(x) + \nabla h(x)\|$ is bounded by $R - \sqrt{\epsilon}$. The small constant $\epsilon$ is used for numerical stability.

## 7.2 Convergence Proof

Following the second Theorem outlined by Défossez et al. (Défossez et al., 2020), we calculate the convergence bound of our algorithm with a dynamic learning rate and loss function.

**Theorem 2.** Given the assumptions from Appendix A.2 and applying Lemma A.3, for all components of the step sizes and gradients, update $\alpha_i$ with the corresponding value from $H_t^S$ and $H_t^T$. Let $\{x_k\}$ be a sequence generated by the optimizer, with $0 < \beta_2 \leq 1$, and $\alpha_i > 0$. For any time step $K$, we have,

$$\mathbb{E}\left[\|\nabla F(x_k)_2^2\|\right] \leq 2R \frac{F(x_0) - f^*}{\lambda \alpha_i K} + C \tag{13}$$

where

$$C = \frac{1}{K}\left(\frac{2\alpha_i R}{\sqrt{1-\beta_2}} + \frac{\alpha_i^2 L}{2(1-\beta_2)}\right)\left(\ln\left(\frac{(1-\beta_2^k)R^2}{(1-\beta_2)\epsilon}\right) - \ln(\beta_2)\right)$$

The detailed proof is provided in Appendix A.4.

CoMMIT adjusts both $\lambda$ and $h(x)$ to balance multimodal learning progress. The parameter $\lambda$, which measures the balance between feature and language learning, remains greater than 1 during training, driven by the balance metric $\tilde{\kappa}$. By avoiding learning oscillations, $\lambda$ can grow even larger, contributing to faster learning. When $\tilde{\kappa} > 1$, the model suffers from insufficient feature learning. CoMMIT reduces the learning rate of the LLM to balance learning, ensuring $\lambda = \frac{\tilde{\kappa}_t + 1}{\tilde{\kappa}_{t-1} + 1} > 1$. Conversely, when $\tilde{\kappa} < 1$, the model suffers from insufficient language learning, ensuring $\lambda = \frac{1/\tilde{\kappa}_t + 1}{1/\tilde{\kappa}_{t-1} + 1} > 1$. Notably, $\nabla h(x)$ is directly added to $\nabla f(x)$ to induce gradient changes, which further contributes to the increase of $\lambda$, resulting in a faster convergence rate.

In this section, we show the proof using Adam as the base optimizer. Due to the reason that CoMMIT does not modify the optimization algorithm itself, the theorem can also be extended to any gradient-based optimization method such as the stochastic gradient descent.

## 8 Experiment

**Experiment setup** We conduct experiments on two non-text modalities, vision and audio, with multiple instruction-tuning downstream tasks: (1) for **Vision**, we fine-tune the pre-trained BLIP-2 (Li et al., 2023), which consists of a vision Q-Former (*i.e.*, the feature encoder) and a backbone OPT-2.7B LLM (Zhang et al., 2022). We evaluate on three visual question-answering tasks: TextVQA

(Singh et al., 2019), IconQA (Lu et al., 2021), and A-OKVQA (Schwenk et al., 2022), which focus on text recognition and reasoning, knowledge-intensive QA, and abstract diagram understanding, respectively; (2) for **Audio**, we leverage the SALMONN (Tang et al., 2023a) model, which extracts both speech and audio features from waveforms and composes these low-level features by a learnable audio Q-Former structure (*i.e.*, the feature encoder). The audio tokens generated by the audio Q-Former are prefixed to the language instruction tokens, which are then input to the backbone Vicuna-7B LLM (Chiang et al., 2023). We evaluate one audio question-answering task and two audio captioning tasks: ClothoAQA (Lipping et al., 2022), MACS (Morato & Mesaros, 2021), and SDD (Manco et al., 2023), which focus respectively on crowdsourced audio question-answering, acoustic scene captioning, and text-to-music generation.

We follow the common instruction tuning diagram (Dai et al., 2024; Tang et al., 2023a; Huang et al., 2023a), where the parameters of backbone LLMs are finetuned with LoRAs (Hu et al., 2021) and the feature encoders are finetuned directly. We set the learning rate to $1e-4$ for all the feature encoders and backbone LLMs in our baseline methods **Constant LR** (Dai et al., 2024; Tang et al., 2023a), **Feature CD**, **Language CD** (Wright, 2015). For Feature CD, we first update the feature encoder until its weights stabilize, then update the backbone LLMs. For Language CD, the process is reversed, with the LLMs being trained first. We also use $1e-4$ as the base learning rates for our CoMMIT variants. There are two CoMMIT variants: **CoMMIT** and **CoMMIT-CLR**. **CoMMIT** is out proposed method in this paper, while **CoMMIT-CLR** is an ablation on **CoMMIT**, without the regularization in Section 6.2.

**Improved Learning Efficiency in MLLM Instruction Tuning.** We evaluate the learning efficiency of the proposed methods CoMMIT-CLR and CoMMIT in comparison with Constant LR in Figure 4 and 5. For visual question-answering tasks in Figure 4, we observe that CoMMIT-CLR and CoMMIT are able to accelerate the instruction tuning of BLIP-2 in the early stage. This is especially in the task of IconQA which is out-of-domain in the pretraining of BLIP-2 (Dai et al., 2024). Specifically, IconQA requires regional-level and spatial visual understanding that are different from pre-trained tasks (Chen et al., 2023). In addition, CoMMIT-CLR and CoMMIT can achieve lower training losses compared with Constant LR. These validates that CoMMIT improves the training efficiency.

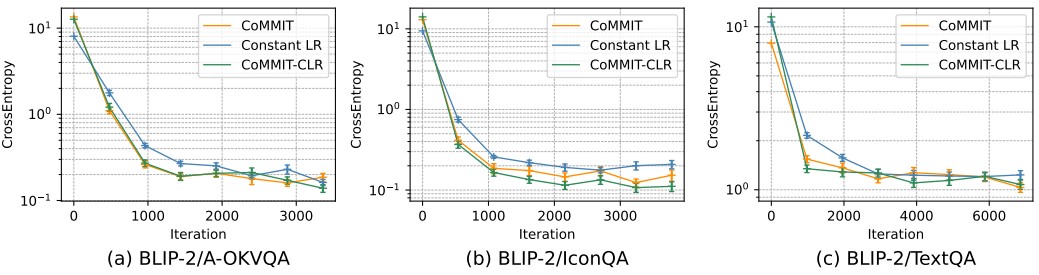

Figure 4: Instruction-tuning learning curves of BLIP-2 on three vision-based downstream tasks.

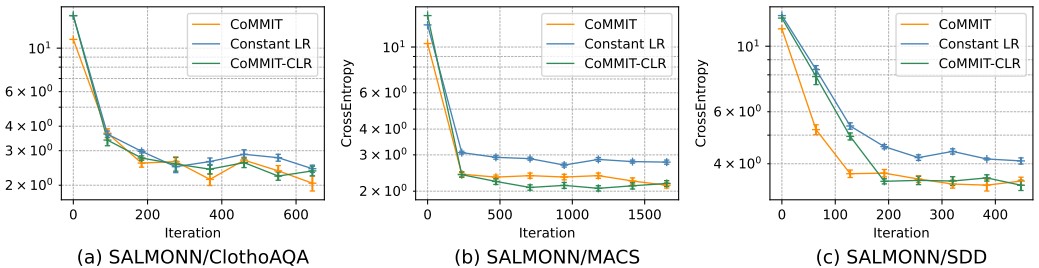

Figure 5: Instruction-tuning learning curves of SALMONN on three audio-based downstream tasks.

Similar to the vision-based tasks, we can find in Figure 5 that CoMMIT-CLR and CoMMIT can also converge to lower loss values in audio tasks. Specifically, we observe that the CoMMIT-CLR and CoMMIT can achieve better accelerations on the audio captioning tasks of MACS and SDD,

compared to training on the audio question-answering task of ClothoAQA. Since audio captioning tasks need more adaptation in MLLMs to generate relatively longer context and align the generation distribution with specific tasks, the coordinated learning rate scheduling method in Section 6.1 can more dynamically adjust the learning rate for less learned components at each model update step. In addition, we show that the proposed loss regularization method adopted in CoMMIT can actively promote the difference in MLLM's generation distribution between optimization steps, which can better benefit tasks, such as audio captioning, that require the model to generate longer contexts.

| Model | Task | Constant LR | Feature CD | Language CD | CoMMIT CLR | CoMMIT |
|-------|------|-------------|------------|-------------|------------|--------|
| **BLIP-2** | A-OKVQA | 54.06 | 57.99 | 49.87 | 60.44 | **64.37** |
| | IconQA | 37.16 | 35.48 | 34.47 | **39.09** | 38.65 |
| | TextVQA | 26.48 | 18.00 | 19.44 | 27.66 | **28.12** |
| **SALMONN** | ClothoAQA | 42.49 | 45.80 | 38.52 | **52.86** | 50.55 |
| | MACS | 24.60 | 22.41 | 23.64 | 23.81 | **25.06** |
| | SDD | 15.10 | 5.70 | **15.74** | 15.07 | 15.33 |
| **InternVL2** | A-OKVQA | 76.59 | 73.19 | 79.47 | 78.00 | **80.52** |
| | IconQA | 80.94 | **83.20** | 81.60 | 80.85 | 82.87 |
| | TextVQA | 65.22 | 65.60 | 65.08 | 65.18 | **67.00** |
| **LLaVA-1.5** | A-OKVQA | 79.20 | 77.64 | 76.94 | 77.82 | **79.55** |
| | IconQA | 64.09 | 64.16 | 58.17 | 65.78 | **69.60** |
| | TextVQA | 41.98 | 43.34 | **49.32** | 47.80 | 49.30 |

Table 1: Instruction tuning results for four MLLMs: BLIP-2, SALMONN, InternVL2-8B, and LLaVA-1.5-7B. These are pre-trained LLMs in vision and audio respectively. For questions-answering tasks like A-OKVQA, IconQA, TextVQA, and ClothoAQA, we report the accuracy score of the generated answers. For audio captioning tasks (MACS and SDD), we report the Rouge-L metric that compares the generated caption with candidate captions. We highlight the best method in bold font for each downstream task of instruction tuning.

**Improved Downstream Performance across Modalities.** In Table 1, We evaluate the performance of the proposed methods CoMMIT-CLR and CoMMIT, comparing with three baselines Constant LR, Feature CD, and Language CD. Among the three baselines, we observe that coordinate gradient descend methods have the most improvement compared to the constant learning rate methods that show significant learning tendencies towards a certain modality (*e.g.*, Language CD in SDD, and Feature CD in A-OKVQA and ClothoAQA). However, since such learning balance varies in downstream tasks, coordinate descent methods cannot consistently improve MLLM instruction tuning, while arbitrarily inclining towards only a certain modality can result in inferior model performance (*e.g.*, Feature CD in SDD and Language CD in A-OKVQA).

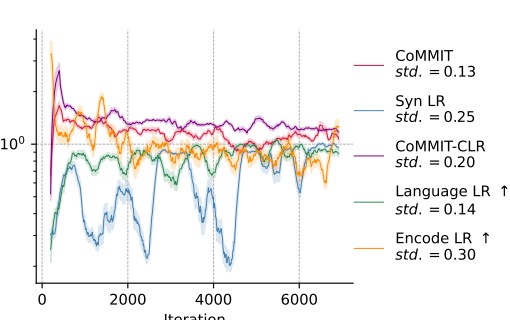

Figure 6: Learning curves of the multimodal learning balance coefficient $\kappa_t$ for multiple methods. In addition to the learning curve, we also report the standard deviation of $\kappa_t$ of each method.

Different from the fixed learning tendency which needs to be predetermined by coordinate descent methods, the proposed coordinated learning rate scheduling method can dynamically adapt learning rates for multimodal components and balance the multimodal joint training. With better coordinated multimodal learning, CoMMIT-CLR and CoMMIT consistently improve Constant LR across modalities and downstream tasks. In addition, the proposed regularization in CoMMIT can promote larger step sizes in gradient descent, which enlarges differences in in the generated output distributions between differen time steps. This prevents learning from being stuck at local optima, which can be

especially beneficial for modality-specific captioning tasks whose optimization space can be relatively larger than question-answering tasks.

**Balanced Multimodal Learning.** In Figure 6, we evaluate the stability of our proposed CoMMIT and CoMMIT-CLR in comparison with the three learning rate scheduling methods described in Section 5. We report with the BLIP-2 (Li et al., 2023) backbone model on the task of TextVQA (Singh et al., 2019). It can be observed that both CoMMIT and CoMMIT-CLR can stabilize multimodal learning with smaller standard deviations of $\kappa_t$ over time. Though Language LR $\uparrow$ also yields high stability on $\kappa$, such learning rate adjustment method suffers

| Method | A-OKVQA | TextVQA | IconVQA |
|--------|---------|---------|---------|
| **LR=1e-5** | 50.30 | 27.58 | 35.45 |
| **LR=1e-4** | 54.06 | 26.48 | 37.16 |
| **LR=1e-3** | 45.24 | 20.60 | 34.93 |
| **CoMMIT** | **64.37** | **28.12** | **38.65** |

Table 2: Comparison on A-OKVQA, TextVQA, and Icon-VQA with BLIP-2 backbone model. Baselines are Constant LR that direct fine tune the backbone model with various learning rate. Comparative, our CoMMIT dynamically adjusts its learning rate.

from the problems of imbalanced training between the feature encoder and LLM as described in Section 5.2, which would potentially cause insufficient training on the feature encoder and worse performance of instruction tuning. Comparing CoMMIT and CoMMIT-CLR, we can observe that CoMMIT achieves more balanced learning with the value of $\kappa_t$ closer to 1, while demonstrating relatively milder learning oscillation with less variant $\kappa_t$ during training. Such better stability in CoMMIT can be benefited by the loss regularization in Section 6.2, which encourages generation distribution change in MLLMs conditioned on the learning progress of the feature encoder and language model. Accompanied by the loss regularization, the learning balance coefficient $\kappa_t$, which is calculated based on generation distributions, can be more accurately estimated and the coordinated learning rate scheduler can more effectively adapt the optimization process.

Note that SynLR that has oscillated value of $\kappa_t$ is actually our baseline Constant LR. In Figure 4 and 5, it is shown that Constant LR generally has lower rate of training (descent on loss values) at the earlier stage. These are consistent with Hypothesis 4.1 that the oscillation problem can slow the training of MLLM in instruction tuning.

**Comparing various learning rates.** In Table 2, we compare CoMMIT with results of constant LR with different learning rate. We report the results on tasks of A-OKVQA, TextVQA, and Icon-VQA with BLIP-2 backbone model. We can observe that our proposed CoMMIT outperforms the Constant LR baselines with significant margin. In addition, we can also find that there is no fixed value of learning rate that consistently yields the best performance for Constant LR, while out proposed CoMMIT is able to dynamically adjust its learning rate. These resuts demonstrate the necessity and effectiveness of dynamic learning rate adjustment for balanced learning between the feature encoder and LLM in multimodal instruction tuning.

## 9 CONCLUSION

In this work, we address the challenge of imbalanced learning between the feature encoder and the backbone LMM during MLLM instruction tuning. Through theoretical analysis and empirical observations, we uncovered how this imbalance can lead to insufficient learning and the oscillation problem. To mitigate these challenges, we proposed **CoMMIT**, a novel approach that dynamically coordinates the learning rates of the feature encoder and LLM backbone. Our **CoMMIT** also included regularization on the gradient gradients that promotes training sufficiency. Our theoretical and empirical analyses demonstrate that **CoMMIT** improves the overall learning efficiency. Experiments across multiple vision and audio downstream instruction tuning tasks illustrate that the training with **CoMMIT** for MLLMs is more effective compared to baselines.

Our work has the potential limitations as follows: (i) the MLLMs which we focus on have a similar architecture design that is composed of a feature encoder and a backbone LLM for reasoning; (ii) the proposed method only focuses on MLLM instruction tuning but may not be directly generalized to MLLM pre-training.

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

## A APPENDIX

### A.1 LEARNING BALANCE IN MULTIMODAL JOINT TRAINING

*Proof.* According to Lipschitz continuity in cross-entropy loss function (Mao et al., 2023), there exists a sequence of $T_t$ and $S_t$ during MLLM instruction tuning, where multimodal components are jointly trained. Given the two metric spaces, $(\mathbb{R}, l_2)$ of the cross-entropy losses and $(H, d)$ of the generation distributions, there exists $0 < \gamma < 1$ such that, at each optimization step $t$,

$$\left\| L\left(P_{X_t,S_t}\right) - L\left(P_{X_{t+1},S_{t+1}}\right) \right\|_2 \leq \gamma d\left(P_{X_{t+1},S_{t+1}} || P_{X_t,S_t}\right), \tag{14}$$

where the metric $d$ measures the change in the prediction distribution $P_{X_t,S_t} \in H$ as the multimodal components $X$ and $S$ are updated. Based on the triangle inequality in metric space, a joint step of multimodal learning is bounded by the combination of two components' separate step forward,

$$d\left(P_{X_{t+1},S_{t+1}} || P_{X_t,S_t}\right) \leq d\left(P_{X_{t+1},S_t} || P_{X_t,S_t}\right) + d\left(P_{X_t,S_{t+1}} || P_{X_t,S_t}\right), \tag{15}$$

where the first term represents the change due to updating the $X$-component while keeping $S$ fixed, and the second term represents the change due to updating the $S$-component.

Then we can derive the multimodal gradient estimated bounds based on MLLM's generative performance in its metric space $d$ shown in the Theorem 1. $\qquad\square$

### A.2 NECESSARY ASSUMPTIONS

We state the necessary assumptions (Bertsekas et al., 2003) commonly used when analyzing the convergence of stochastic algorithms for non-convex problems:

**Assumption 1.** *The minimum value of $f(x)$ is lower-bounded,*

$$\forall x \in \mathbb{R}^d, \ f^* = \min f(x).$$

**Assumption 2.** *The gradient of the non-convex objective function $f$ is L-Liptchitz continuous (Nesterov, 2013). Then $\forall x, y \in \mathbb{R}^d$, the following inequality holds,*

$$f(y) \leq f(x) + \nabla f(x)^T(y - x) + \frac{L}{2}\|x - y\|_2^2.$$

### A.3 CONTROLLING DEVIATION FROM DESCENT DIRECTION

Following the first Lemma outlined by Défossez et al. (Défossez et al., 2020), where the expected update direction can positively correlate with the gradient (Sashank et al., 2018), we aim to control the deviation from the descent direction to enhance convergence.

**Lemma 1.** For all $k \in \mathbb{N}^*$ and $R \geq \|\nabla f(x) + \nabla h(x)\| + \sqrt{\epsilon}$, the gradient update follows a descent direction,

$$\mathbb{E}_{k-1}\left[\nabla_i F(x_{k-1}) \frac{\lambda \nabla_i f_k(x_{k-1}) + \nabla_i h_k(x_{k-1})}{\sqrt{\epsilon + v_{k,i}}}\right] - \frac{\lambda(\nabla_i F(x_{k-1}))^2}{2\sqrt{\epsilon + \tilde{v}_{k,i}}}$$

$$\geq \frac{\nabla_i F(x_{k-1}) \nabla_i h_k(x_{k-1})}{2\sqrt{\epsilon + \tilde{v}_{k,i}}} - 2R\mathbb{E}_{k-1}\left[\frac{(\lambda \nabla_i f_k(x_{k-1}) + \nabla_i h_k(x_{k-1}))^2}{\epsilon + v_{k,i}}\right]. \tag{16}$$

*Proof.* Denote $F = \nabla_i F(x_{k-1})$, $f = \lambda \nabla_i f_k(x_{k-1})$, $h = \nabla_i h_k(x_{k-1})$, and $\tilde{v}_{k,i} = \beta_2 v_{k-1,i} + \mathbb{E}_{k-1}\left[(\lambda \nabla_i f_k(x_{k-1}) + \nabla_i h_k(x_{k-1}))^2\right]$, we get:

$$\mathbb{E}_{k-1}\left[\frac{F(f+h)}{\sqrt{\epsilon + v_{k,i}}}\right] = \mathbb{E}_{k-1}\left[\frac{F(f+h)}{\sqrt{\epsilon + \tilde{v}_{k,i}}}\right] + \mathbb{E}_{k-1}\left[F(f+h)\left(\frac{1}{\sqrt{\epsilon + v_{k,i}}} - \frac{1}{\sqrt{\epsilon + \tilde{v}_{k,i}}}\right)\right] \quad (17)$$

We know that $g$ and $\tilde{v}_{k,i}$ are independent given $f_1, f_2, \cdots, f_{n-1}$. $h$ and $\tilde{v}_{k,i}$ are also independent based on our settings which do not affect the momentum, we have,

$$\mathbb{E}_{k-1}\left[\frac{F(f+h)}{\sqrt{\epsilon + \tilde{v}_{k,i}}}\right] = \frac{\lambda F^2}{\sqrt{\epsilon + \tilde{v}_{k,i}}} + \frac{Fh}{\sqrt{\epsilon + \tilde{v}_{k,i}}} \quad (18)$$

The only thing we need to do is control the deviation of the second term in Eq.( 17). Applying Cauchy-Schwarz (Steele, 2004),

$$RHS = F(f+h)\frac{\mathbb{E}_{k-1}\left[(f+h)^2\right] - (f+h)^2}{\sqrt{\epsilon + v_{k,i}}\sqrt{\epsilon + \tilde{v}_{k,i}}(\sqrt{\epsilon + v_{k,i}} + \sqrt{\epsilon + \tilde{v}_{k,i}})}$$

$$\leq F(f+h)\frac{\mathbb{E}_{k-1}\left[(f+h)^2\right]}{\sqrt{\epsilon + v_{k,i}}(\epsilon + \tilde{v}_{k,i})} + F(f+h)\frac{(f+h)^2}{\sqrt{\epsilon + v_{k,i}}(\epsilon + \tilde{v}_{k,i})}. \quad (19)$$

By applying the inequality $ab \leq \frac{1}{2\lambda}b^2 + \frac{\lambda}{2}a^2$ with $\lambda = \frac{\sqrt{\epsilon + \tilde{v}_{k,i}}}{2}$, a $= \frac{F}{\sqrt{\epsilon + \tilde{v}_{k,i}}}$, and $b = \frac{(f+h)\mathbb{E}_{k-1}\left[(f+h)^2\right]}{\sqrt{\epsilon + \tilde{v}_{k,i}}\sqrt{\epsilon + v_{k,i}}}$, the conditional expectation of the first term in Eq.( 19) can be bounded as,

$$\mathbb{E}_{k-1}\left[F(f+h)\frac{\mathbb{E}_{k-1}\left[(f+h)^2\right]}{\sqrt{\epsilon + v_{k,i}}(\epsilon + \tilde{v}_{k,i})}\right] \leq \mathbb{E}_{k-1}\left[\frac{F^2}{4\sqrt{\epsilon + \tilde{v}_{k,i}}} + \frac{(f+h)^2\mathbb{E}_{k-1}\left[(f+h)^2\right]^2}{\sqrt{\epsilon + \tilde{v}_{k,i}}^3(\epsilon + v_{k,i})}\right]$$

$$\leq \frac{F^2}{4\sqrt{\epsilon + \tilde{v}_{k,i}}} + \mathbb{E}_{k-1}\left[\frac{(f+h)^2\mathbb{E}_{k-1}\left[(f+h)^2\right]}{\sqrt{\epsilon + \tilde{v}_{k,i}}(\epsilon + v_{k,i})}\right]$$

$$\leq \frac{F^2}{4\sqrt{\epsilon + \tilde{v}_{k,i}}} + R\mathbb{E}_{k-1}\left[\frac{(f+h)^2}{\epsilon + v_{k,i}}\right], \quad (20)$$

with respect to the fact that $\epsilon + \tilde{v}_{k,i} \geq \mathbb{E}_{k-1}\left[(f+h)^2\right]$ and $\mathbb{E}_{k-1}\left[(f+h)^2\right] \leq R$.

Similarly, applying the inequality $ab \leq \frac{\lambda}{2}a^2 + \frac{1}{2\lambda}b^2$ with $\lambda = \frac{\sqrt{\epsilon + \tilde{v}_{k,i}}}{2\mathbb{E}_{k-1}\left[(f+h)^2\right]}$, $a = \frac{F(f+h)}{\sqrt{\epsilon + \tilde{v}_{k,i}}}$, and $b = \frac{(f+h)^2}{\epsilon + v_{k,i}}$, the conditional expectation of the second term in Eq.( 19) can be bounded as,

$$\mathbb{E}_{k-1}\left[F\frac{(f+h)^2(f+h)}{\sqrt{\epsilon + v_{k,i}}(\epsilon + \tilde{v}_{k,i})}\right] \leq \mathbb{E}_{k-1}\left[\frac{F^2}{4\sqrt{\epsilon + \tilde{v}_{k,i}}}\frac{(f+h)^2}{\mathbb{E}_{k-1}\left[(f+h)^2\right]} + \frac{\mathbb{E}_{k-1}\left[(f+h)^2\right]}{\sqrt{\epsilon + \tilde{v}_{k,i}}}\frac{(f+h)^4}{(\epsilon + v_{k,i})^2}\right]$$

$$\leq \frac{F^2}{4\sqrt{\epsilon + \tilde{v}_{k,i}}} + \mathbb{E}_{k-1}\left[\frac{\mathbb{E}_{k-1}\left[(f+h)^2\right]}{\sqrt{\epsilon + \tilde{v}_{k,i}}}\frac{(f+h)^2}{(\epsilon + v_{k,i})}\right]$$

$$\leq \frac{F^2}{4\sqrt{\epsilon + \tilde{v}_{k,i}}} + R\mathbb{E}_{k-1}\left[\frac{(f+h)^2}{\epsilon + v_{k,i}}\right], \quad (21)$$

given again $\mathbb{E}_{k-1}\left[(f+h)^2\right] \leq R$.

Putting inequalities (20) and (21) back into (19) gives,

$$\mathbb{E}_{k-1}\left[F(f+h)\left(\frac{1}{\sqrt{\epsilon + v_{k,i}}} - \frac{1}{\sqrt{\epsilon + \tilde{v}_{k,i}}}\right)\right] \leq \frac{F^2}{2\sqrt{\epsilon + \tilde{v}_{k,i}}} + 2R\mathbb{E}_{k-1}\left[\frac{(f+h)^2}{\epsilon + v_{k,i}}\right] \quad (22)$$

And, therefore, adding Eq.(22) and Eq.(18) into Eq.(17) finishes the proof. $\qquad\square$

### A.4 PROOF OF CONVERGENCE

In this section, we prove the theorem 2.

*Proof.* Given $\alpha_k = \alpha\sqrt{\frac{1-\beta_2^k}{1-\beta_2}}$, we apply the Assumption 2 and get,

$$F(x_k) \le F(x_{k-1}) - \alpha_k \nabla F(x_{k-1})^T u_k + \frac{\alpha_k^2 L}{2}\|u_k\|_2^2. \tag{23}$$

Since we define the bound $R \ge \|\nabla f(x) + \nabla h(x)\| + \sqrt{\epsilon}$, it follows that $\sqrt{\epsilon + \tilde{v}_{k,i}} \le R\sqrt{\sum_{j=0}^{n-1} \beta_2^j}$. By applying this inequality, we obtain,

$$\alpha_k \left( \frac{(\lambda\nabla_i F(x_{k-1}))^2}{2\sqrt{\epsilon + \tilde{v}_{k,i}}} + \frac{\nabla_i F(x_{k-1})\nabla_i h_k(x_{k-1})}{2\sqrt{\epsilon + \tilde{v}_{k,i}}} \right)$$
$$\ge \alpha \left( \frac{(\lambda\nabla_i F(x_{k-1}))^2}{2R} + \frac{\nabla_i F(x_{k-1})\nabla_i h_k(x_{k-1})}{2R} \right). \tag{24}$$

By taking the conditional expectation, we apply Eq.( 24) to Lemma 1 to derive results from Eq.( 23),

$$\mathbb{E}_{k-1}\left[F(x_K)\right] \le \mathbb{E}_{k-1}\left[F(x_{k-1})\right] - \frac{\alpha\lambda}{2R}\|\nabla F(x_k)_2^2\|$$
$$- \frac{\alpha}{2R}(\nabla F(x_k)^T \nabla h(x_k)) + \left(2\alpha_k R + \frac{\alpha_k^2 L}{2}\right)\mathbb{E}\left[\|u_k\|_2^2\right] \tag{25}$$

Summing the previous inequality over all $k$ and taking the full expectation with respect to the fact that $\alpha \ge \alpha_k\sqrt{1-\beta_2}$. By applying Lemma 5.2 from Défossez et al. (Défossez et al., 2020), we get the final bound,

$$\mathbb{E}\left[\|\nabla F(x_k)_2^2\|\right] \le 2R\frac{F(x_0) - f^*}{\alpha(1+\lambda)K} + \left(\frac{2\alpha R}{\sqrt{1-\beta_2}} + \frac{\alpha^2 L}{2(1-\beta_2)}\right)\left(\frac{1}{K}\ln\left(\frac{(1-\beta_2^n)R^2}{(1-\beta_2)\epsilon}\right) - ln(\beta_2)\right) \tag{26}$$

$\square$

### A.5 COMPUTATION RESOURCES

Our model is trained on 4 A100 GPUs with 40GB memory. The average training time is about 8 hours.

## B ANALYSIS ON THE COMPARISON OF NORMALIZED LEARNING GRADIENTS

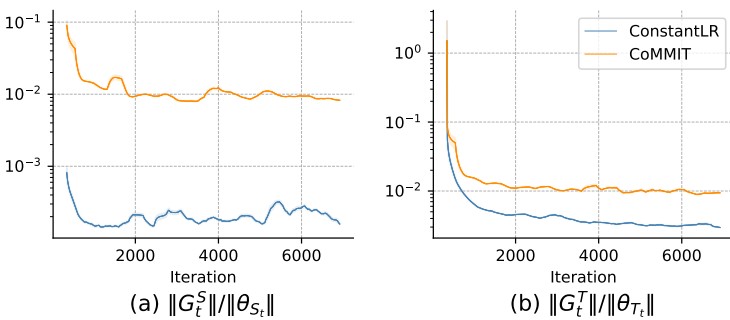

(a) $\|G_t^S\|/\|\theta_{S_t}\|$          (b) $\|G_t^T\|/\|\theta_{T_t}\|$

Figure 7: The learning curves of normalized learning gradient $\|G_t^S\|/\|\theta_t^S\|$ and $\|G_t^X\|/\|\theta_t^X\|$ for the feature encoder and language model respectively.

