# OpenReview forum: "CoMMIT: Coordinated Instruction Tuning for Multimodal Large Language Models"
_ICLR.cc/2025/Conference — Submitted to ICLR 2025_

### Official Review · Reviewer_W36A · 2024-10-27

**Soundness:** 2
**Presentation:** 2
**Contribution:** 2
**Rating:** 3
**Confidence:** 3

**Summary:**

This work analyzes instruction tuning in multimodal large language models (MLLMs) from both theoretical and empirical perspectives, and finds unbalanced learning between the feature encoder and the LLM can cause problems of oscillation learning and insufficient training with diminishing learning gradients. To alleviate this, they propose a multimodal balance coefficient to measure the learning balance, and introduce an auxiliary regularisation on the gradient. Experiments on four multimodal LLMs show the proposed method outperforms the baselines.

**Strengths:**

1. This paper analyzes instruction tuning in multimodal LLMs and finds unbalanced learning between the feature encoder and the LLM can cause problems of oscillation learning and insufficient training with diminishing learning gradients.
2. They propose a multimodal balance coefficient as well as a dynamic learning scheduler to alleviate oscillation learning and insufficient training.
3. Empirical results on multiple downstream tasks in vision and audio modalities show the proposed method CoMMIT outperforms the baselines.

**Weaknesses:**

1. The contribution looks limited. The proposed method seems to be hard to follow, since it is customized for multimodal instruction tuning.
2. The experiments are not solid enough to confirm the effectiveness of CoMMIT. This paper might consider more recent MLLMs with different architectures.
3. The presentation can be improved. For example, baselines such as Constant LR, Feature CD and Language CD should be briefly explained, to avoid any confusion.

**Questions:**

- Although this paper is about multimodal instruction tuning, I am curious about whether the findings and proposed method can be generalised to other post-training schemas such supervised fine-tuning (SFT) and direct preference optimization (DPO). If so, do authors have any experimental results under SFT and DPO settings?
- Why don't select some different LLM backbones such as Cambrian-1, MiniCPM-V-2.6 and Qwen2-VL?

---

> ### Author Response · Authors · 2024-11-23
> **Rebuttal to Reviewer W36A**
>
> Thank you for your valuable feedback and the time you have spent reviewing our work. We address the concerns raised and provide answers to your questions accordingly.
>
>
> **Responses to Weakness 1**
>
> The setting in our paper addresses a prevailing and significant challenge. Instruction tuning has emerged as a critical technique and a primary method for adapting MLLMs to downstream tasks in recent works [1-8], especially given the computational constraints of pre-training MLLMs from scratch.
>
> The major challenge in MLLM instruction tuning lies in effectively aligning the feature encoder for downstream tasks with the text features, ensuring more relevant modality-specific information. Our paper proposes a new method that addresses this challenge by dynamically balancing multimodal learning progress by dynamically coordinating learning rates on the feature encoder and LLM.
>
> **Responses to Weakness 2 and Question 2**
>
> As discussed in Section 4, our paper focuses on achieving a balance between the feature encoder and the backbone LLM during MLLM instruction tuning.
> Our method is designed to be easily implemented and compatible with a wide range of MLLMs during fine-tuning.
> To demonstrate the generalizability of our approach, we use popular and widely adopted backbone model architectures in state-of-the-art MLLMs. For instance, the backbone models BLIP-2 [1] and LLaVA [2] used in our paper are both highly cited.
>
> Moreover, based on popular leaderboards [9-11], LLaVA [2] achieves the best performance among open-sourced models on MLLM-Bench [9]. Additionally, InternVL outperforms Qwen on benchmarks such as MLVU [10] and MMMU [11]. To further support the flexibility of our method, we include SALMONN[8] for audio tasks, demonstrating its adaptability across different modalities.
> Given the popularity and diversity of the backbone models we selected, we believe they are sufficient to prove the generalizability and effectiveness of our method.
>
> **Responses to Weakness 3**
>
> We have already explained all the baselines in detail in lines 389–397 of the paper. To summarize:
>
> 1. **Constant LR**: This is the standard supervised fine-tuning (SFT) approach. Both the feature encoder and backbone LLM are fine-tuned using LoRAs with a fixed learning rate of $1e^{-4}$.
> 2. **Feature CD**: The feature encoder is updated first until its weights stabilize, followed by training the backbone LLM with the same learning rate.
> 3. **Language CD**: The reverse of Feature CD, where the backbone LLM is trained first, and then the feature encoder is updated.
> 4. **CoMMIT Variants**: We also evaluate CoMMIT (our proposed method) and CoMMIT-CLR (an ablation of CoMMIT without the regularization term), both of which use an initial learning rate of $1e^{-4}$.
> \end{itemize}
>
> **Responses to Question 1**
>
> We have already included the SFT method as a baseline in our experiments, which is Constant LR. As previously mentioned, our method is designed to address the challenge of achieving synergy through cooperative learning across different modalities. This enables LLMs to adapt their reasoning abilities to downstream tasks while feature encoders adjust to provide more relevant modality-specific information. By leveraging this unique architecture of MLLMs, our approach can be seen as a more effective solution compared to traditional SFT methods.
> DPO falls outside the scope of this work. However, as highlighted earlier, our setting is generalized enough to be broadly applicable to multimodal instruction tuning.
>
>
> [1] Li, Junnan, et al. "Blip-2: Bootstrapping language-image pre-training with frozen image encoders and large language models." International conference on machine learning. PMLR, 2023.
>
> [2] Liu, Haotian, et al. "Visual instruction tuning." Advances in neural information processing systems 36 (2024).
>
> [3] Chen, Zhe, et al. "Internvl: Scaling up vision foundation models and aligning for generic visual-linguistic tasks." CVPR. 2024.
>
> [4] Zhang, Renrui et al. “LLaMA-Adapter: Efficient Fine-tuning of Large Language Models with Zero-initialized Attention.” International Conference on Learning Representations (2024).
>
> [5] Gao, Peng, et al. "Llama-adapter v2: Parameter-efficient visual instruction model." arXiv:2304.15010.
>
> [6] Lee, Byung-Kwan, et al. "Collavo: Crayon large language and vision model." arXiv:2402.11248.
>
> [7] Han, Jiaming, et al. "Imagebind-llm: Multi-modality instruction tuning." arXiv:2309.03905.
>
> [8] Tang, Changli, et al. "SALMONN: Towards Generic Hearing Abilities for Large Language Models." ICLR. 2024.
>
> [9] Ge, Wentao, Shunian Chen, and G. Hardy Chen. "MLLM-Bench: evaluating multimodal LLMs with per-sample criteria." arXiv:2311.13951.
>
> [10] Zhou, Junjie, et al. "MLVU: A Comprehensive Benchmark for Multi-Task Long Video Understanding." arXiv:2406.04264.
>
> [11] Yue, Xiang, et al. "Mmmu: A massive multi-discipline multimodal understanding and reasoning benchmark for expert agi." CVPR. 2024.

---

> > ### Author Response · Authors · 2024-11-27
> > **Looking forward to the discussion**
> >
> > Dear Reviewer W36A,
> >
> > We sincerely appreciate the time and effort you have dedicated to reviewing our work, especially given your undoubtedly busy schedule. We are eager to understand whether our reply has effectively addressed your concerns and to learn if there are any additional questions or points you would like to discuss.
> >
> > Thank you once again for your thoughtful consideration, and we look forward to any further feedback you may have.
> >
> > Best regards,
> >
> > The Authors

---

### Official Review · Reviewer_6Pft · 2024-11-02

**Soundness:** 2
**Presentation:** 2
**Contribution:** 2
**Rating:** 5
**Confidence:** 3

**Summary:**

The paper addresses unbalanced learning between the LLM and feature encoder in multimodal instruction tuning, leading to issues like oscillation and insufficient training. It proposes a Multimodal Balance Coefficient and a dynamic learning scheduler to coordinate learning, alongside an auxiliary regularization to improve training efficiency. The proposed techniques are architecture-agnostic and show improved performance across multiple tasks and models.

**Strengths:**

**Combination of Theory and Empirical Evidence**: The proposed theoretical framework is combined with empirical observations, revealing potential issues of learning imbalance and providing deep insights.

**Dynamic Coordination of Learning**: CoMMIT dynamically adjusts the learning rates of the feature encoder and LLM to effectively balance multimodal learning progress, avoiding oscillations and insufficient training.

**Broad Applicability**: The proposed method can be applied to different optimizers and various LLMs, demonstrating strong general applicability.

**Weaknesses:**

**Limited Generalizability**: It is unclear whether the observed phenomenon is universal, as the authors only used BLIP-2 model and TextVQ dataset in their empirical studies, raising concerns about generalizability.

**Lack of Novel Model Architecture**: The paper primarily proposes a parameter tuning method. A new model architecture would have been more impactful, rather than just dynamically adjusting learning rates.

**Questions:**

- Using more models and more data in the empirical analysis would make the findings more convincing.

- The authors used three VQA datasets for testing in Table 2. To my knowledge, multimodal large models have many downstream tasks, more evaluation datasets should be included.

---

> ### Author Response · Authors · 2024-11-23
> **Rebuttal to Reviewer 6Pft**
>
> Thank you for your valuable feedback and the time you have spent reviewing our work. We address the concerns raised and provide answers to your questions accordingly.
>
>
> **Response to Weakness 1 and Question 1**
>
> In Figure 4 and Figure 5, we empirically studied the learning curves of ConstantLR and CoMMIT on multiple datasets, A-OKVQA, IconQA, TextVQA, ClothoAQA, MACS, and SDD, with multiple backbone models, including LLaVA-1.5, BLIP-2, SALMONN, and InternVL2 in both the vision and audio domains.
> In Line 399 - 405, We demonstrated the learning inefficiency problem in ConstantLR and showed that CoMMIT can accelerate the learning process and converge to lower estimation errors.
> In addition, we also included the comparative study in Table 1 of four different backbone MLLMs in multiple datasets, showing consistent improvement of CoMMIT.
>
> To provide more empirical results on the study of the learning oscillation problem,
> we include an analysis of the standard deviations $\kappa$ for LLaVA-1.5 and InternVL on multiple datasets.
> Based on the empirical results, we can observe consistent improvement of CoMMIT on the learning oscillation problem by reducing the variance of learning curves.
>
> | LLaVA-1.5  |   A-OKVQA |   TextVQA |   IconQA |
> |:-----------|----------:|----------:|---------:|
> | CoMMIT     |    0.2576 |    0.3719 |   0.1941 |
> | ConstantLR |    0.3361 |    0.6267 |   0.1519 |
>
> | InternVL   |   A-OKVQA |   TextVQA |   IconQA |
> |:-----------|----------:|----------:|---------:|
> | CoMMIT     |    0.3232 |    0.2434 |   0.2969 |
> | ConstantLR |    0.3448 |    0.6397 |   0.3286 |
>
>
> **Response to Weakness 2 and  Question 2**
>
> We would like to emphasize that one major contribution of our work (as claimed in two contributions at the end of the introduction section) contributes a novel theoretical framework that addresses the learning imbalance problem in MLLM instruction tuning, offering insights into improving optimization through a dynamic learning rate adjustment and a loss regularization term, which fundamentally enhances instruction tuning theory. As discussed in Section 4, our paper focuses on achieving a balance between the feature encoder and the backbone LLM during MLLM instruction tuning.
> Our method is designed to be easily implemented and compatible with a wide range of MLLMs during fine-tuning.
>
> We validate our approach across multiple datasets, A-OKVQA, IconQA, TextVQA, ClothoAQA, MACS, and SDD, which include Visual Question-answering, Optical Character Recognition, Audio Question-answering, and Audio Captioning from diverse domains, including vision and audio tasks. We also evaluate four MLLMs, LLaVA-1.5, BLIP-2, SALMONN, and InternVL2,  demonstrating CoMMIT's  generalization ability and outperforming or matching baselines in widely adopted evaluation protocols. The evaluation in Table 2 is only to justify that no fixed value of learning rate consistently yields the best performance for Constant LR, while our proposed CoMMIT can dynamically adjust its learning rate (as discussed in Line 520 -523).
>
> Our use of datasets and evaluation protocols aligns with recent literature [a,b,c,d,e,f,g,h],
> and we contribute to the growing body of work by addressing optimization challenges rather than proposing another model architecture.
>
> [a] Wang, Sheng, et al. "PRoLoRA: Partial Rotation Empowers More Parameter-Efficient LoRA." arXiv preprint arXiv:2402.16902 (2024).
>
> [b] Jie, Shibo, et al. "Memory-Space Visual Prompting for Efficient Vision-Language Fine-Tuning." Forty-first International Conference on Machine Learning.
>
> [c] Panos, Aristeidis, et al. "Imperfect Vision Encoders: Efficient and Robust Tuning for Vision-Language Models." arXiv preprint arXiv:2407.16526 (2024).
>
> [d] Zhu, Didi, et al. "Model Tailor: Mitigating Catastrophic Forgetting in Multi-modal Large Language Models." Forty-first International Conference on Machine Learning.
>
> [e] He, Jinghan, et al. "Continual instruction tuning for large multimodal models." arXiv preprint arXiv:2311.16206 (2023).
>
> [f] Zhang, Renrui et al. “LLaMA-Adapter: Efficient Fine-tuning of Large Language Models with Zero-initialized Attention.” International Conference on Learning Representations (2024).
>
> [g] Gao, Peng, et al. "Llama-adapter v2: Parameter-efficient visual instruction model." arXiv preprint arXiv:2304.15010 (2023).
>
> [h] Li, Yifan, et al. "Facial Affective Behavior Analysis with Instruction Tuning." arXiv preprint arXiv:2404.05052 (2024).

---

> > ### Comment · Reviewer_6Pft · 2024-11-26
> >
> > I appreciate the author's efforts. From my perspective, I am not very excited about this work and I feel that the contribution is limited, but I am still willing to offer a reward for your rebuttal.

---

### Official Review · Reviewer_MzCX · 2024-11-04

**Soundness:** 3
**Presentation:** 3
**Contribution:** 3
**Rating:** 6
**Confidence:** 4

**Summary:**

- The paper introduces CoMMIT, a novel method for multimodal instruction tuning that dynamically coordinates the learning rates of the multimodal components and employs an auxiliary loss for gradient regularization.
- It establishes a theoretical framework to identify and analyze learning imbalances in multimodal large language model (MLLM) instruction tuning and provides a convergence rate analysis based on this framework.
- Experiments on multiple downstream tasks and modalities demonstrate that CoMMIT improves both convergence rate and effectiveness in MLLM instruction tuning.

**Strengths:**

- The paper identifies the phenomenon of unbalanced learning between the feature encoder and the LLM in the MLLM instruction tuning, which can cause diminishing learning gradients and often lead to sub-optimal results.
    - It introduces a quantitative measure for evaluating learning balance and proposes a coordinated learning rate scheduler with auxiliary loss regularization, effectively coordinating the learning of multimodal components.

**Weaknesses:**

- MLLMs typically comprise a feature encoder, an LLM, and a multimodal projector (e.g., the q-former in BLIP2), the paper does not discuss the role of the multimodal projector in the proposed method. It is unclear if the projector is considered part of the feature encoder, and if so, the rationale behind this choice is not explained.
- Although the paper demonstrates faster convergence of the proposed method, it lacks empirical comparisons in terms of training time efficiency.
- The evaluation of CoMMIT focuses solely on fine-tuning performance. It is unclear if the proposed method could also lead to better performance in the zero-shot setting.
- It would be better if the paper could also include a comparison of normalized learning gradients (as in Figure 3) for the proposed CoMMIT.

**Questions:**

- In Section 5.2, why does using a large learning rate for the feature encoder result in gradient diminishing in the feature encoder S , as shown in Figure 3a?
- Have the authors experimented the proposed method with different optimizers? Can the advantage brought by the proposed method be generalized to different optimizers.
- Typos in L271

---

> ### Author Response · Authors · 2024-11-23
> **Rebuttal to Reviewer MzCX**
>
> Thank you for your valuable feedback and the time you have spent reviewing our work. We address the concerns raised and provide answers to your questions accordingly.
>
>
> **Response to Weakness 1**
>
> Yes, the multimodal projector is treated as part of the encoder $S$.
> Specifically, when training with BLIP2, we freeze the image encoder and finetune the q-former $S$ and LLM $X$. This is reasonable since there should still exist the insufficient and oscillation problem when learning with these two modules. Our results on BLIP2 show that our CoMMIT is generic to cooperative learning between different modules in MLLM instruction tuning.
>
> **Response to Weakness 2**
>
> Since our major contribution is the multimodal learning theory in MLLM instruction fine-tuning, we follow previous works in model optimization [a,b,c] and report the learning curve comparisons between different learning methods in Table 4 and Table 5. Empirically, since CoMMIT does not change the fine-tuning structure of MLLMs, we are expecting a similar computation complexity. In addition, we analyze the computational cost for calculating  $k_t$  and the associated regularization terms as  $T = 2N_S + N_t + 2N$,
> where  $N_S$  and  $N_t$  are significantly smaller than  $N$  due to parameter-efficient tuning, making the additional cost marginal relative to the overall training.
>
> **Response to Weakness 3**
>
> As explained in our previous response, since our major contribution is the multimodal learning theory in MLLM instruction fine-tuning, we follow the instruction tuning evaluation protocol [d,e,f,g] to evaluate on fine-tuning performance.
>
> **Response to Weakness 4**
>
> Thanks for the suggestion. We added the suggested comparison in Appendix B (Figure 7), which compares the normalized learning gradients between ConstantLR and CoMMIT. We can observe that CoMMIT could significantly improve the gradient diminishing issue.
>
>
> **Response to Question 1**
>
> The two methods evaluated in Figure 3, Encoder LR ↑ and Language LR ↑, indicate a setting of imbalanced learning rates during training. As stated in Observation 5.2 (Line 265 - 267), we observe that imbalanced learning that inclines toward X or S can result in gradient diminishing and inferior training performance.
>
> **Response to Question 2**
>
> Yes, as mentioned in line 370, our method does not modify the optimization algorithm itself but focuses on updating the learning rate. Therefore, it can be extended to any gradient-based optimization method. We have tested it with stochastic gradient descent (SGD), and our method consistently outperforms the baseline, demonstrating its generalizability.
>
> **Response to Question 3**
>
> We apologize for the typo. In L271, "Such In such cases" should be "In such cases".
>
>
> [a]. Iiduka, Hideaki. "Appropriate learning rates of adaptive learning rate optimization algorithms for training deep neural networks." IEEE Transactions on Cybernetics 52.12 (2021): 13250-13261.
>
> [b]. Liu, Liyuan, et al. "On the variance of the adaptive learning rate and beyond." arXiv preprint arXiv:1908.03265 (2019).
>
> [c]. Na, Gyoung S. "Efficient learning rate adaptation based on hierarchical optimization approach." Neural Networks 150 (2022): 326-335.
>
> [d] Yin, Zhenfei, et al. "Lamm: Language-assisted multi-modal instruction-tuning dataset, framework, and benchmark." \textit{Advances in Neural Information Processing Systems} 36 (2024).
>
> [e] Chen, Chi, et al. "Position-enhanced visual instruction tuning for multimodal large language models." \textit{arXiv preprint arXiv:2308.13437} (2023).
>
> [f] Li, Zou, Ning Pang, and Xiang Zhao. "Instruction Tuning Large Language Models for Multimodal Relation Extraction Using LoRA." \textit{International Conference on Web Information Systems and Applications}. Singapore: Springer Nature Singapore, 2024.
>
> [g] Panagopoulou, Artemis, et al. "X-instructblip: A framework for aligning x-modal instruction-aware representations to llms and emergent cross-modal reasoning." \textit{arXiv preprint arXiv:2311.18799} (2023).

---

> > ### Author Response · Authors · 2024-11-27
> > **Looking forward to the discussion**
> >
> > Dear Reviewer MzCX,
> >
> > We sincerely appreciate the time and effort you have dedicated to reviewing our work, especially given your undoubtedly busy schedule. We are eager to understand whether our reply has effectively addressed your concerns and to learn if there are any additional questions or points you would like to discuss.
> >
> > Thank you once again for your thoughtful consideration, and we look forward to any further feedback you may have.
> >
> > Best regards,
> >
> > The Authors

---

> > > ### Comment · Reviewer_MzCX · 2024-11-27
> > >
> > > Thank you for the response! It has addressed some of my concerns. However, I do believe one of the key advantages of instruction tuning lies in its ability to enhance zero-shot performance on novel tasks. Therefore, it is important to show if the proposed method could also lead to better performance in the zero-shot setting. With that being said, I think the work demonstrates good performance in terms of multimodal fine-tuning, and I would like to maintain my score.

---

### Official Review · Reviewer_zTnv · 2024-11-05

**Soundness:** 2
**Presentation:** 2
**Contribution:** 3
**Rating:** 5
**Confidence:** 4

**Summary:**

The paper addresses the issue of imbalanced learning between the feature encoder and the LLM in multimodal instruction tuning, which leads to insufficient training and oscillation problems. To alleviate the issue, the authors propose a new training strategy with a dynamic learning scheduler and gradient regularization to balance and enhance learning. Empirical results demonstrate improved convergence and performance across various multimodal tasks with various MLLM.

**Strengths:**

•	The problem discussed in the paper, i.e., the imbalanced training in MLLMs, is interesting and meaningful for multimodal learning and broader research communities.

•	The paper proposes CoMMIT, a coordinated learning rate scheduler that effectively balances the training of the feature encoder and LLM.

•	Through theoretical analysis, the paper demonstrates that CoMMIT leads to faster convergence and can be generalized across various optimizers.

•	Empirical results across various downstream multi-modal tasks prove that CoMMIT is both effective and adaptable to different MLLM architectures.

**Weaknesses:**

•	The conclusions of this paper may not be generalizable due to its limited experiment setup. It seems the authors only investigated the setting of finetuning with LoRAs. But LoRAs finetuning can be very different full finetuning. So the generalizability of the approach and findings under this setup is questionable.

•	The paper does not have a clear definition of “learning insufficiency”. In Hypothesis 4.2, the authors mention “imbalanced learning can cause insufficient learning problem”, but do not establish clear criteria or metrics that differentiate sufficient from insufficient learning. Providing a more rigorous definition (e.g., quantifiable definition or threshold) for “learning insufficiency” could strengthen the theoretical and empirical claims.

•	The empirical experiments do not directly demonstrate that CoMMIT resolves the oscillation and insufficient learning issues. While the learning curves and instruction tuning results on MLLMs show overall improvements, they lack in-depth analysis that proves the specific problems are addressed.

•	The experiment setup is not clearly illustrated and lacks many important details. For example, in Section 8, the authors did not clearly state what instruction tuning datasets they are using, what’s the size of the dataset. They also didn’t provide the setup for InternVL2 and LLaVA-1.5.

**Questions:**

•	How does the variation of Multimodal Balance Coefficient κ during training correlate with model performance and training stability? It would be helpful if you could add detailed quantitive analysis or case studies to show κ’s impact.

•	Although the paper discusses gradient regularization to prevent diminishing gradients, can you provide a more intuitive and in-depth analysis on how the regularization can affect gradients behaviors? For example, more detailed gradient visualization such as gradient norms would be helpful to demonstrate the effectiveness.

•	There are some typos in the draft. For instance, in lines 24-25, “which potentially prevents enables a more …” seems to be a grammar error.

---

> ### Author Response · Authors · 2024-11-23
> **Rebuttal to Reviewer zTnv (Part 1/2)**
>
> Thank you for your valuable feedback and the time you have spent reviewing our work. We address the concerns raised and provide answers to your questions accordingly.
>
>
> **More empirical results** of the standard deviations of $\kappa$ for LLaVA-1.5 and InternVL on multiple datasets. Our proposed CoMMIT induces $\kappa$ with smaller variances, improving training stability.
>
> | LLaVA-1.5  |   A-OKVQA |   TextVQA |   IconQA |
> |:-----------|----------:|----------:|---------:|
> | CoMMIT     |    0.2576 |    0.3719 |   0.1941 |
> | ConstantLR |    0.3361 |    0.6267 |   0.1519 |
>
> | InternVL   |   A-OKVQA |   TextVQA |   IconQA |
> |:-----------|----------:|----------:|---------:|
> | CoMMIT     |    0.3232 |    0.2434 |   0.2969 |
> | ConstantLR |    0.3448 |    0.6397 |   0.3286 |
>
>
> **Response to Weakness 1**
>
> Parameter-efficient fine-tuning with LoRA in MLLM instruction tuning is a technique adopted in a wide range of works [a,b,c,d,e,f,g,h,i,j,k]. How to balance the learning between these two components can be a general challenge that concerns different model backbones [f,g,h,i], and modalities [i,j,k]. We will highlight our contribution on such points to avoid misunderstanding.
>
> **Response to Weakness 2**
>
> Quantitatively, the insufficient learning corresponds to $\kappa>>1$ or $\kappa\rightarrow 0$ (Hypothesis 4.2). When these happen, the learning would be primarily attributed to the updates on only one of the encoders or LLM, resulting in the other module (LLM or encoder) being learned insufficiently. On the contrary, sufficient learning refers to $\kappa$ close to 1 with both the encoder and LLM being involved in the learning dynamics.
>
> **Response to Weakness 3**
>
> From the Table above (along with Figure 6), our proposed CoMMIT resolves the oscillation problem via inducing less variant $\kappa$, i.e., the training is more stable without oscillating between the encoder and LLM.
>
> Additionally, it can be observed from Figure 6 that the $\kappa$ from CoMMIT is closer to 1, suggesting both the encoder and LLM are involved in the training dynamics. This improves the learning sufficiency as compared to biasing on either the encoder ($\kappa\rightarrow 0$) or LLM ($\kappa>>1$), e.g., Encode LR $\uparrow$ and Language LR $\uparrow$ in Figure 2.
>
> From a theoretical perspective, as shown in Equation 13, applying COMMIT results in a better bound on the norm of the gradient.
> Specifically, $\lambda$ is always greater than 1, leading to an upper bound on the gradient norm that is smaller than the bound achieved by the original Adam algorithm.
> This indicates more efficient and sufficient learning.
>
> **Response to Weakness 4**
>
> In Section 8, we provide the information (on Line 378 - 380) of the instruction tuning datasets for vision tasks, TextVQA (34K training, 5K test), A-OKVQA (17K training, 6K test), and IconQA (18K training, 6K test). For audio tasks, we explain the datasets (on Line 385 - 387) ClothoAQA (21K training, 8K test), MACS (3K training, 393 test), and SDD (1K training, 746 test). We follow the original data split provided by the individual dataset.
> We follow the original prompt template for InternVL2 and LLaVA-1.5. We enable the instruction tuning by only calculating the loss on the response tokens. We will add such information in our paper to enable better implementation details.
>
> **Response to Question 1**
>
> $\kappa$ of smaller variance suggests the learning is not oscillating between the encoder and LLM, thus is an indicator of training stability. In the table above, our proposed CoMMIT improves the training stability with a much lower variance in $\kappa$ compared to ConstantLR. As a result, CoMMIT constantly yields better performance than ConstantLR  (in Table 1).
>
> **Response to Question 2**
>
> Thanks for the suggestion. We added the suggested comparison in Appendix B (Figure 7) corresponding to our findings in Section 5.2 (Figure 3), which compares the normalized learning gradients between ConstantLR and CoMMIT. We can observe that CoMMIT could significantly improve the gradient diminishing issue.
>
> **Response to Question 3**
>
> We apologize for the typo. in lines 24-25 the sentence should be "which potentially enables a more accurate estimation ...".
>
> [a] Wang, Luping, et al. "Parameter-Efficient Fine-Tuning in Large Models: A Survey of Methodologies." \textit{arXiv preprint arXiv:2410.19878} (2024).
>
> [b] Zhou, Xiongtao, et al. "An Empirical Study on Parameter-Efficient Fine-Tuning for MultiModal Large Language Models." \textit{arXiv preprint arXiv:2406.05130} (2024).
>
> [c] He, Jinlong, et al. "Pefomed: Parameter efficient fine-tuning on multimodal large language models for medical visual question answering." \textit{arXiv preprint arXiv:2401.02797} (2024).
>
> [d] Li, Zou, Ning Pang, and Xiang Zhao. "Instruction Tuning Large Language Models for Multimodal Relation Extraction Using LoRA." WWW, 2024.

---

> ### Author Response · Authors · 2024-11-23
> **Rebuttal to Reviewer zTnv (Part 2/2)**
>
> [e] Jin, Yizhang, et al. "Efficient multimodal large language models: A survey." \textit{arXiv preprint arXiv:2405.10739} (2024).
>
> [f] Chen, Shaoxiang, Zequn Jie, and Lin Ma. "Llava-mole: Sparse mixture of lora experts for mitigating data conflicts in instruction finetuning mllms." \textit{arXiv preprint arXiv:2401.16160} (2024).
>
> [g] Xue, Le, et al. "xgen-mm (blip-3): A family of open large multimodal models." \textit{arXiv preprint arXiv:2408.08872} (2024).
>
> [h] Gao, Peng, et al. "Llama-adapter v2: Parameter-efficient visual instruction model." \textit{arXiv preprint arXiv:2304.15010} (2023).
>
> [i] Tang, Changli, et al. "Salmonn: Towards generic hearing abilities for large language models." \textit{arXiv preprint arXiv:2310.13289} (2023).
>
> [j] Ye, Qilang, et al. "Cat: Enhancing multimodal large language model to answer questions in dynamic audio-visual scenarios." \textit{European Conference on Computer Vision}. Springer, Cham, 2025.
>
> [k] Sagare, Shivprasad, et al. "Audio-visual training for improved grounding in video-text LLMs." \textit{arXiv preprint arXiv:2407.15046} (2024).

---

> > ### Author Response · Authors · 2024-11-27
> > **Looking forward to the discussion**
> >
> > Dear Reviewer zTnv,
> >
> > We sincerely appreciate the time and effort you have dedicated to reviewing our work, especially given your undoubtedly busy schedule. We are eager to understand whether our reply has effectively addressed your concerns and to learn if there are any additional questions or points you would like to discuss.
> >
> > Thank you once again for your thoughtful consideration, and we look forward to any further feedback you may have.
> >
> > Best regards,
> >
> > The Authors

---

> > > ### Comment · Reviewer_zTnv · 2024-11-28
> > > **Thank you for the response**
> > >
> > > I appreciate the detailed response from the authors. Most of my questions and concerns are addressed. However, I still think the authors should demonstrate if the conclusions could be generalized to other experiment setups. I decide to increase the score to 5.

---

### Official Review · Reviewer_HDWE · 2024-11-11

**Soundness:** 2
**Presentation:** 1
**Contribution:** 3
**Rating:** 5
**Confidence:** 4

**Summary:**

This paper focus on the balance of learning between vision encoder and llm in the context of visual instruction tuning.  The imbalanced learning is caused by two problems: (1) insufficient learning and (2) oscillation of gradient. To address these two problem, this paper proposes CoMMIT consisting a coordinated learning rate scheduler and regularization in gradient descent.

The paper (1) defines the Multimodal balance coefficient (k) as the ration between two KL divergence and proved that k accounts for the upper bound of the gradient for llm and vision encoder.
(2) proposes regularization to avoid gradient diminishing problem.

The results show that proposed CoMMIT and CoMMIT-CLR can accelerate the convergence of the losses on two modalities (image and audio) and help the models to achieve lower losses.

**Strengths:**

1. this paper points out an important and interesting problem in multimodal training, i.e., the training balance between the vision encoder and the LLM.

2. This paper provide both empirical results and theoretical analysis to support their proposed Multimodal balance coefficient.

**Weaknesses:**

1. writing needs to be significantly improved. many typos and grammar errors, making the paper hard to follow:
L24-25 prevents enables
L200  observe the dyanmics of κt is different
L203 bewteen
L212 - 213
L 283 show case,  problems that signifies

2.  observations in 5.1 and 5.2 need further explanations. (see Questions 4)

3. What are the benefits of the proposed regularization in terms of empirical results? such as convergence speed of the losses or model's performance? If not, it's hard to justify the usefulness of this method.

4. Missing analysis and discussion:
(1) How often do you need to compute k_t in order to get an accurate estimation? can you discuss the optimal updating interval of k_t?
(2) what is the latency caused by computing k_t?

**Questions:**

1. in equation (6), what do you mean by logits?

2. in appendix A.1, line 716, what is T_t?

3. line 721 "prediction distribution" of what?

4. why increasing the lr of encoder causes the K_t going to 1 in Figure 2 (c)? shouldn't it go to zero?

5. what is unsupervised instruction tuning on L 311. maybe provide a reference?

6. what is \tilde{X_{\theta}^x_t} in equation (9)?

7. Line 322-323, what is the relationship between N and K?

8. equation (25), what does F(x_k)^2_2 mean?

---

> ### Author Response · Authors · 2024-11-23
> **Rebuttal to Reviewer HDWE**
>
> Thank you for your valuable feedback and the time you have spent reviewing our work. We address the concerns raised and provide answers to your questions accordingly.
>
>
> **Response to Weakness 1**
>
> Thanks for pointing them out. We have corrected these errors in our current draft.
>
> **Response to Weakness 2 and Question 4**
>
> In Figure 2 (c), the $\kappa_t$ "Encode LR $\uparrow$" is actually close to 0 at the earlier steps as compared to later when it converges to near 1. This demonstrates the imbalanced learning that inclines toward the encoder when $\kappa_t\rightarrow0$ in Hypothesis 4.2.
>
> We contend that it is not increasing the lr of the encoder that causes $\kappa_t$ going to 1 in later training steps, since all three different lr setups in Figure 2 (c) end up  $\kappa_t$ converging to close to 1. In explaining this, we cautiously speculate that the numerator and denominator  $\kappa$ are both more and more resulting from the randomness of similar scales when close to converge. This causes $\kappa_t$ close to 1 in the later stage of training irrespective of the learning rate setup.
>
> **Response to Weakness 3**
>
> We conducted a comprehensive ablation study of the proposed regularization in Table 1.
> As discussed in Line 484 - 487, the proposed regularization in CoMMIT can promote larger step sizes in gradient descent, which enlarges differences in the generated output distributions between different time steps.
>
> **Response to Weakness 4**
>
> We compute  $k_t$  every  $K = 10$  training step, which balances the need for accurate estimation with computational efficiency,
> as this interval ensures that the values remain relevant without incurring excessive overhead.
> The computational cost for calculating  $k_t$  and the associated regularization terms is  $T = 2N_S + N_t + 2N$,
> where  $N_S$  and  $N_t$  are significantly smaller than  $N$  due to parameter-efficient tuning, making the additional cost marginal relative to the overall training.
> Given the reduced parameter sizes and the periodic computation of  $k_t$, the latency caused by this operation is negligible compared to the dominant costs of training the entire MLLM.
> Therefore, the choice of the updating interval $K$ is to optimize the trade-off between maintaining accurate updates to  $k_t$  and minimizing computational impact.
>
>
> **Response to Question 1**
>
> Logits refer to the unnormalized outputs of the model's final layer, consistent with their definition in deep learning. These are the raw scores produced before applying any activation function.
>
> **Response to Question 2**
>
> We apologize for the confusion. The term $T_t$ should actually refer to $X_t$, the pre-trained language model $X$ at the t-th step of training.
>
> **Response to Question 3**
>
> As defined on Line 137, it represents the prediction distribution of the generated response when the multimodal components are jointly updated.
>
> **Response to Question 5**
>
> On L 311, we refer to the learning process only relying on instruction without access to the responses. Such regularization can be applicable to unsupervised learning in representation learning [a], domain generalization [b,c], and domain adaptation [d,e].
>
>
> **Response to Question 6**
>
> Sorry for the confusion. On L 311, the notation should be $\tilde{X}_{\theta^x_t}=T(X;\theta^x_t)$. We will add the definition to improve readability.
>
> **Response to Question 7**
>
> We apologize for the typo. $N$ should actually be $K$, representing the number of component functions.
>
> **Response to Question 8**
>
> $\|\nabla F(x_k)^2_2\|$ represents the squared norm of the gradient. Ideally, at a global minimum, this value should be 0. In optimization, proving that this term is bounded by a finite value is a key step to demonstrate convergence.
>
>
> [a] Xie, Baao, et al. "Graph-based Unsupervised Disentangled Representation Learning via Multimodal Large Language Models." \textit{arXiv preprint arXiv:2407.18999} (2024).
>
> [b] Cheng, De, et al. "Disentangled Prompt Representation for Domain Generalization." \textit{Proceedings of the IEEE/CVF Conference on Computer Vision and Pattern Recognition}. 2024.
>
> [c] Choi, Juhwan, et al. "VolDoGer: LLM-assisted Datasets for Domain Generalization in Vision-Language Tasks." \textit{arXiv preprint arXiv:2407.19795} (2024).
>
> [d] Zhang, Huanyu, et al. "LogoRA: Local-Global Representation Alignment for Robust Time Series Classification." \textit{IEEE Transactions on Knowledge and Data Engineering} (2024).
>
> [e] Chen, Dongjie, et al. "Empowering Source-Free Domain Adaptation with MLLM-driven Curriculum Learning." \textit{arXiv preprint arXiv:2405.18376} (2024).

---

> > ### Author Response · Authors · 2024-11-27
> > **Looking forward to the discussion**
> >
> > Dear Reviewer HDWE,
> >
> > We sincerely appreciate the time and effort you have dedicated to reviewing our work, especially given your undoubtedly busy schedule. We are eager to understand whether our reply has effectively addressed your concerns and to learn if there are any additional questions or points you would like to discuss.
> >
> > Thank you once again for your thoughtful consideration, and we look forward to any further feedback you may have.
> >
> > Best regards,
> >
> > The Authors

---

> > > ### Comment · Reviewer_HDWE · 2024-11-28
> > >
> > > Thanks for the clarification made by the authors to address my questions and thanks for correcting typos and grammar errors. I still have a few questions and hope to discuss.
> > >
> > > (1) For weakness 4 about the value of K, I understand it's a trade off between minimizing the computation cost and accuracy. But how do you decide this number K=10? Since you mentioned that this won't impose much computational cost, why not set K to 1, so you have the most accurate learning rate. or K=10 is sufficient and decreasing K won't bring you much benefit? I think this should be considered since it can bring more empirical insights.
> > >
> > > (2) For Q8, shouldn't the standard way of writing the norm of gradient be $|\nabla f(x_k)|^2_2$ ?
> > >
> > > (3) For Q4, can you elaborate more on "In explaining this, we cautiously speculate that the numerator and denominator
> > >  are both more and more resulting from the randomness of similar scales when close to converge."?
> > > what do you mean by "randomness of similar scales"? Why does this happen?

---

### Meta-Review · Area_Chair_kCGF · 2024-12-15

**Metareview:**

This paper proposes CoMMIT, a method for improving multimodal instruction tuning in large language models by addressing the problem of imbalanced learning between feature encoders and language models. While the concept of balancing learning rates through a multimodal balance coefficient and auxiliary gradient regularization is interesting, the execution and evaluation of the work have several critical shortcomings. The method's generalizability is questionable due to its limited evaluation on diverse models and datasets. Additionally, the paper lacks clarity in its explanations and the empirical evidence provided is insufficient to substantiate the claims about convergence improvements and training stability. These limitations outweigh the strengths of the proposed method, and therefore the paper is recommended for rejection.

**Additional Comments On Reviewer Discussion:**

Reviewers raised concerns about the limited scope of experiments, lack of clarity in presenting the method, and insufficient generalizability of the proposed approach. While the authors addressed some issues during the rebuttal, such as additional experiments and clarifications, the responses failed to resolve the fundamental concerns about scalability and robustness. The consensus among reviewers reflects the need for more rigorous evaluation and clearer articulation of the contributions, leading to the decision to reject.

---

### Decision · Program_Chairs · 2025-01-22

Reject